# Endogenous *n*-Alkanes in Vegetable Oils: Validation of a Rapid Offline SPE-GC-FID Method, Comparison with Online LC-GC-FID and Potential for Olive Oil Quality Control

**DOI:** 10.3390/molecules28114393

**Published:** 2023-05-28

**Authors:** Ana Srbinovska, Paolo Lucci, Chiara Conchione, Laura Barp, Sabrina Moret

**Affiliations:** 1Department of Agri-Food, Environmental and Animal Sciences, University of Udine, Via Sondrio 2A, 33100 Udine, Italy; srbinovska.ana@spes.uniud.it (A.S.); chiara.conchione@uniud.it (C.C.); laura.barp@uniud.it (L.B.); 2Department of Agricultural, Food and Environmental Sciences, Polytechnic University of Marche, 60131 Ancona, Italy; p.lucci@univpm.it

**Keywords:** extra-virgin olive oil, avocado oil, sunflower oil, *n*-alkanes, offline SPE-GC-FID, online HPLC-GC-FID

## Abstract

The potential of endogenous *n*-alkane profiling for the assessment of extra virgin olive oils (EVOO) adulteration (blends with cheaper vegetable oils) has been studied by relatively few authors. Analytical methods used for this purpose often involve tedious and solvent-intensive sample preparation prior to analytical determination, making them unattractive. A rapid and solvent-sparing offline solid phase extraction (SPE) gas chromatography (GC) flame ionization detection (FID) method for the determination of endogenous *n*-alkanes in vegetable oils was, therefore, optimized and validated. The optimized method demonstrated good performance characteristics in terms of linearity (R^2^ > 0.999), recovery (on average 94%), and repeatability (residual standard deviation, RSD < 11.9%). The results were comparable to those obtained with online high-performance liquid chromatography (HPLC)-GC- FID ( RSD < 5.1%). As an example of an application to prove the potentiality of endogenous *n*-alkanes in revealing frauds, the data set obtained from 16 EVOO, 9 avocado oils (AVO), and 13 sunflower oils (SFO), purchased from the market, was subjected to statistical analysis and principal component analysis. Two powerful indices, namely (*n*-C_29_ + *n*-C_31_)/(*n*-C_25_ + *n*-C_26_) and *n*-C_29_/*n*-C_25_, were found to reveal the addition of 2% SFO in EVOO and 5% AVO in EVOO, respectively. Further studies are needed to confirm the validity of these promising indices.

## 1. Introduction

Due to its high economic value and its unique sensory, compositional and nutritional characteristics, extra virgin olive oil (EVOO) is at high risk of fraud, one of the most common being the marketing of blends of EVOO with cheaper vegetable oils. Different vegetable oils having a similar fatty acid and/or sterol composition such as sunflower (SFO) and hazelnut oil (HZO), or of lower price (e.g., palm oil, PLO, and avocado oil, AVO) are the most common adulterants [1].

Among the possible analytical tools to uncover these frauds, *n*-alkane profiling has great potential, but to date, it has rarely been applied.

*n*-Alkanes are aliphatic saturated hydrocarbons made up of carbon and hydrogen, with a linear carbon chain following the empiric formula C_n_H_2n+2_. They are nonpolar organic compounds naturally present in the unsaponifiable fraction of vegetable oils, probably deriving from the decarboxylation of long-chain fatty acids [2].

Vegetable oils contain *n*-alkanes from *n*-C_21_ to *n*-C_35_ with a predominance of odd-numbered carbon chain terms [3], which, after proper sample preparation, can be quantified by gas chromatography (GC) coupled to a flame ionization detector (FID). Although endogenous *n*-alkanes co-elute with saturated mineral oil hydrocarbons (MOSH), fortunately, they are well distinguishable from petrogenic-derived hydrocarbons due to the balanced distribution between even and odd carbon number hydrocarbons, i.e., an even to odd “carbon preference index” (CPI) close to 1, often accompanied by the presence of one or more “humps” consisting of a large number of iso- and cyclo-alkanes [4] and a negligible amount of *n*-alkanes in the *n*-C_25_–C_35_ range. Thus, even in the presence of MOSH contamination, the typical profile of *n*-alkanes is well recognizable and can be used to characterize vegetable oils.

Several researchers studied the *n*-alkane composition of EVOO demonstrating their ability in discriminating oils of different varieties [5,6]. Koprivnjak et al. (2005) used linear discriminant analysis (LDA) to correctly assign olives of three different cultivars from Croatia (Leccino, Buza, Bjelica) harvested during four consecutive years at three different stages of ripening [7]. Other parameters may influence the *n*-alkane profile masking differences due to the variety. The influence of geographical origin on the *n*-alkane composition of EVOO has been reported by several authors [2,8,9,10]. Webster et al. (2000) reported the *n*-alkane profile of 20 authentic EVOO and 20 retail EVOO from three different origins (Italy, Spain and Greece): Greek EVOO had higher average levels of total *n*-alkanes (152.5 mg/kg), followed by Italian (71.7 mg/kg) and Spanish ones (43.6 mg/kg) [2].

The *n*-alkane profile of EVOO can be also influenced by refining and refining conditions [11,12,13], the ripening degree of olives [13,14], and the presence of leaves [9].

Despite the many variables that can influence the qualitative-quantitative composition of *n*-alkanes in EVOO, several studies have shown that the compositional differences among oils from different botanical origins are greater, allowing EVOO to be distinguished from other vegetable oils. McGill et al. (1993) proved that *n*-C_27_, *n*-C_29_, and *n*-C_31_ predominated in all the plant oils evaluated except in EVOO where *n*-C_23_, *n*-C_25_, and *n*-C_27_ were the most significant [11]. Troya et al. (2015) used LDA to distinguish vegetable oils from seven different plant origins [15]. Ratios of the peak areas selected by pairs were used as predictors, the most powerful being *n*-C_21_/*n*-C_25_ and *n*-C_24_/*n*-C_30_. Using the LDA model, all the oils were correctly classified according to their botanical origin, with assignment probabilities higher than 95%.

Analytical determination of *n*-alkanes in vegetable oils has been recently reviewed by Srbinovska et al. (2020) [3]. GC-FID allows for proper *n*-alkanes separation and quantification. Nevertheless, prior to GC analysis a proper sample preparation aimed at eliminating triglycerides is needed. Most of the proposed methods involve a saponification step, followed by unsaponifiable extraction and fractionation on a glass column filled with silica gel. As an alternative, some researchers directly isolated *n*-alkanes from triglycerides using adsorption chromatography on a fat retainer (e.g., silica gel). These methods usually make use of large solvent volumes and require intensive sample preparation. Online high-performance liquid chromatography (HPLC)-GC-FID has been also applied for rapid and solvent-sparing determination of *n*-alkanes in olive oils from different cultivars [7], but such instrumentation is not available in all laboratories.

Given the importance of endogenous *n*-alkanes quantification and profiling for quality control purposes of olive oils, there was a need to develop a simple method for their monitoring in laboratories that do not have dedicated online HPLC-GC-FID instrumentation. The main purpose of the present work was, therefore, to validate a rapid and solvent-sparing offline solid phase extraction (SPE)-GC-FID method as an alternative to other more complex and solvent-consuming methods, or to online HPLC-GC-FID. Finally, after demonstrating the equivalence of the offline and online methods here presented, a number of EVOO (declared 100% Italian), refined SFO, and virgin AVO, purchased from the Italian market, were analyzed. Through the application of chemometrics, the potential of *n*-alkanes in revealing the adulteration of EVOO with small percentages of SFO and AVO was studied for the first time.

## 2. Results and Discussion

### 2.1. Optimization of the Offline SPE-GC-FID Procedure

As reported in Materials and Methods, SPE was carried out on a 1 g silver silica gel cartridge according to the method proposed by Moret et al. (2011) for MOSH determination in vegetable oils [16] with some modifications regarding the elution volume, the internal standard chosen for quantification (*n*-C_20_), and the amount injected into the GC-FID apparatus. The silver silica (10%) not only enables the retention of triglycerides but also has an advantage over activated silica gel due to its greater retention power against interfering olefins that might otherwise co-elute with *n*-alkanes.

To optimize the elution volume, an AVO with a high amount of total *n*-alkanes was loaded on a 1-g silver silica cartridge. Then the cartridge was eluted with *n*-hexane collecting different fractions that were concentrated and injected. The first fraction (0–1.0 mL) corresponded to the cartridge dead volume and was free from endogenous *n*-alkanes. The subsequent 1.0–2.5 mL fraction contained most of the *n*-alkanes. Little amounts of *n*-alkanes were also present in the 2.5–3.0 mL fraction, while the subsequent fraction was completely free of them. Based on these results, it was decided to discharge the first mL of eluate (fraction 0–1 mL, corresponding to the dead volume), and to collect *n*-alkanes in the following two mL (fraction 1.0–3.0 mL). In this way, the fraction of interest containing the *n*-alkanes is eluted in 2 mL of *n*-hexane, which is properly concentrated (to about 700 µL) before analytical determination.

Quantification was performed with the internal standard. For this purpose, *n*-C_20_ was chosen over *n*-C_40_ since, based on repeatability data, it gave a slightly lower residual standard deviation (RSD).

GC conditions were optimized in order to achieve good separation among *n*-alkanes while keeping analysis times short. Good separation and resolution of all peaks were obtained by applying a temperature rate of 20 °C/min, so these conditions were selected for method validation. The same conditions were also adopted for the online method. The possible discrimination between *n*-alkanes was regularly checked and found to be negligible.

### 2.2. Interference by MOSH

The possible presence of MOSH of petrogenic origin in the *n*-alkane fraction can be recognized by the characteristic balanced distribution of even and odd carbon number hydrocarbons and/or the presence of humps consisting of unresolved compounds. Anyway, under the condition chosen, the interference by MOSH was mostly not evident or negligible and even when visible in the chromatographic trace, it did not compromise *n*-alkanes quantification. In fact, for MOSH/MOAH determination according to the method of Menegoz Ursol et al. (2022) [17], 20–50 times higher sample amounts are required.

### 2.3. Method Performance

Method performance was tested on a mixture of seven selected *n*-alkanes (the most represented ones) present at different concentration ranges in different vegetable oils. Table 1 shows the linearity range investigated (expressed in mg/kg of oil), the equation of the regression line, and the coefficient of determination (R^2^) obtained by the least squares method.

The coefficients of determination (R^2^) were all greater than 0.999, demonstrating excellent linearity over the entire ranges of concentration tested.

Method precision was verified by assessing recovery (three replicates, each one injected twice) at three different fortification levels as reported in Section 3.3.4. Since EVOO already contained appreciable amounts of *n*-alkanes, it was deprived of the endogenous *n*-alkanes and spiked with different amounts of the stock *n*-C_23_–C_35_ alkanes mixture (all even numbered). The removal step (described in Section 3.3.4) was effective, and only small traces of *n*-alkanes remained after this treatment, which were detracted from the fortified samples.

Figure 1 reports, as an example, the calibration lines of 4 (*n*-C_23_, *n*-C_25_, *n*-C_29_, *n*-C_33_) of the seven *n*-alkanes. The amount spiked (expressed in mg/kg of oil), approximately covered the concentration range usually found in EVOO for every single *n*-alkane tested and ranged from 0.2 to 4.2 mg/kg for *n*-C_23_, from 0.3 to 6.7 mg/kg for *n*-C_25_, from 2.1 to 20.4 mg/kg for *n*-C_29_, and from 0.2 to 4.2 mg/kg for *n*-C_33_.

Recoveries, calculated as the percentage ratio of the area obtained from the spiked sample (red points in Figure 1) and the expected area extrapolated from the calibration line, ranged from 85 to 107% (on average 94%).

Repeatability, assessed by analyzing the same EVOO sample (with a low amount of *n*-alkanes) six times within the same week, gave RSD lower than 11.9% for all *n*-alkanes except for *n*-C_34_ (23%).

### 2.4. Comparison with Online HPLC-GC-FID

Online HPLC-GC, when available, represents the best choice for *n*-alkanes determination in vegetable oils, since the sample only needs to be diluted with *n*-hexane and injected into the apparatus, and it allows minimizing solvent consumption and analysis time. The HPLC silica column retains the fat allowing the elution of the *n*-alkane fraction and its transfer to the GC. During the GC run, the HPLC column is backflushed with dichloromethane to remove the fat and reconditioned with *n*-hexane before the next analysis.

The same EVOO sample subjected to repeatability by offline SPE-GC-FID was also used for repeatability test by online HPLC-GC-FID (*n* = 6), giving comparable results (no significant differences at *p* < 0.05, except for *n*-C_34_ present at very low concentration) and lower RSDs (<5.1%).

### 2.5. n-Alkanes in Vegetable Oils of Certain Origin by Offline SPE-GC-FID

The validated SPE-GC-FID method was applied to different vegetable oils, including EVOO, olive oil (OO), SFO, virgin AVO, virgin HZO, and PLO (all of a certain origin). Figure 2 reports the chromatograms obtained.

EVOO and OO contained 53 and 34 mg/kg of total *n*-alkanes, respectively, and showed little difference in the *n*-alkane profile, the more volatile *n*-alkanes up to *n*-C_25_ being less abundant, probably as a result of refining. Greater differences were found between EVOO and the other four vegetable oils. As reported in previous studies [11,18], in all oils, the odd *n*-alkanes between *n*-C_23_ and *n*-C_33_ were the most abundant. EVOO also contained appreciable amounts of *n*-C_24_. The total *n*-alkanes concentration ranged from 7.0 mg/kg in virgin HZO and 14.1 mg/kg in PLO to 96 and 317 mg/kg in refined SFO (high-oleic) and virgin AVO, respectively. *n*-C_23_ was only present in olive oils and AVO samples. These data well agreed with those reported by Troya et al. (2015) [15].

### 2.6. n-Alkanes in Commercial EVOO, AVO and SFO by Online HPLC-GC-FID

To investigate the potentiality of *n*-alkanes in discriminating EVOO from SFO and AVO, and in discovering fraudulent admixtures of EVOO with these two cheaper vegetable oils, a total of 16 EVOO (100% Italian), 13 refined SFO and 9 virgin AVO, purchased from the Italian market were analyzed. For simplicity, and given the comparable method performance, the analysis was performed using online HPLC-GC-FID.

Table 2 reports the content (expressed in mg/kg of oil) of all even *n*-alkanes from *n*-C_23_ to *n*-C_33_. Other minor *n*-alkanes including long-chain *n*-alkanes (present at very low concentrations) up to *n*-C_47_ for AVO, and up to *n*-C_40_ for some SFO, as well as other odd *n*-alkanes were not reported in the table but have been considered for statistical evaluation. Total *n*-alkanes refers to the sum of all detected *n*-alkanes from *n*-C_21_ to *n*-C_47_.

Both total and individual *n*-alkane concentrations can be used to discriminate EVOO from SFO and AVO. From the data reported in Table 2, it can be seen that EVOO samples always had lower total *n*-alkanes than AVO (except for AVO1) and SFO (except for SFO12). We cannot exclude that these two atypical samples, which differed strongly from other samples of the same type, were adulterated, so more robust data could be obtained by analyzing samples of certain origins. Even better results can be obtained by comparing *n*-C_29_ concentrations of EVOO (8.32–14.47 mg/kg) with those of AVO (33.35–232.86 mg/kg) and SFO (33.55–86.15). Anyway, our choice was to investigate relative percentages of *n*-alkanes, which were subjected to statistical analysis.

#### 2.6.1. Relative Percentage of *n*-Alkanes

By using the relative percentages of *n*-alkanes (see graph in Figure 3), it was possible to appreciate some interesting differences in the *n*-alkane profile of EVOO, AVO, and SFO. In particular, it can be seen that *n*-C_29_ represents, together with *n*-C_23_, *n*-C_24_ and *n*-C_25_, a good parameter for discriminating EVOO from both AVO and SFO, while *n*-C_27_ and *n*-C_31_ seem to be more effective for AVO. Figure 3 displays the minimum (dashed line) and the maximum (solid line) percentage of every single *n*-alkane between *n*-C_21_ and *n*-C_36_ for EVOO compared to AVO (A) and SFO (B).

Figure 3 also allows us to appreciate the general *n*-alkane profile of EVOO, SFO and AVO. While EVOO samples have a similar percentage of *n*-C_29_, *n*-C_25_, *n*-C_27_ and *n*-C_31_ (on average around 15%), followed by a lower percentage of *n*-C_23_ (on average 11%), *n*-C_24_ (7%) and *n*-C_33_ (6.8%), in AVO *n*-C_29_ represent on average 51% of total *n*-alkanes, while in SFO the prevailing *n*-alkane is *n*-C_29_ (on average 45%), followed by *n*-C_27_ (12%) and *n*-C_31_(31%).

#### 2.6.2. Statistical Analysis

The entire data set (relative percentage abundance) was subjected to statistical analysis and was imported into the Metaboanalyst software 5.0 for statistical analysis and principal component analysis (PCA).

In general, predominant *n*-alkanes in EVOO were *n*-C_23_, *n*-C_24_, *n*-C_25_, *n*-C_27_, *n*-C_29_, *n*-C_31_ and *n*-C_33_, whereas in SFO and AVO three major alkanes, namely *n*-C_27_, *n*-C_29_ and *n*-C_31_, accounted for more than 85 and 90% of total alkanes, respectively. These results are consistent with previous works [3,9]. Long-chain *n*-alkanes (from *n*-C_38_ to *n*-C_47_), even at low levels (<1%), have been solely observed in SFO. In general, the total *n*-alkanes content in analyzed EVOO samples was half that of SFO and four times less than AVO samples. These results are clearly highlighted in PCA score plots where EVOO samples are completely separated and distributed in different locations when compared to AVO (Figure 4A) and SFO (Figure 4B). PC1 and PC2 explain 91.6% and 97% of the observed variation with AVO and SFO, respectively, mainly caused by differences in the relative percentage levels of *n*-C_23_, *n*-C_24_, *n*-C_25_, *n*-C_29_, and *n*-C_31_ and thus indicating the suitability of *n*-alkane profiles to distinguish EVOO samples from SFO and AVO.

Particularly, analyzing the differences in the *n*-alkane profile between AVO and EVOO samples with log^2^-fold change (Log^2^FC) (Figure 5), the highest x-fold values obtained when comparing the EVOO profile to both SFO (Figure 5A) and AVO (Figure 5B) was observed for *n*-C_29_ and *n*-C_24_. In fact, EVOO had significantly lower levels of *n*-C_29_ than AVO and SFO while having a higher content of *n*-C_24_. Long-chain *n*-alkanes (from *n*-C_36_ to *n*-C_39_ for AVO and from *n*-C_36_ to *n*-C_47_ for SFO) levels were also significantly higher compared to EVOO. However, their low amounts (<1%) in both AVO and SFO prevent their use as markers for the identification of illegal mixtures of EVOO with AVO or SFO.

In light of this, in order to identify potential indicators able to reveal illegal EVOO blends, the ratio between the relative percentage of *n*-C_29_ and *n*-C_24_ has been initially investigated.

Unfortunately, considering the theoretical changes of the *n*-C_29_/*n*-C_24_ ratio in mixtures of EVOO with SFO and AVO at different percentages, only quite a high level (>30%) of AVO or SFO in EVOO could be detected. Therefore, other indices were evaluated, based on the *t*-test results reported in Figure 6.

Statistical significance in the level of *n*-alkanes (*p* < 0.05) of EVOO and AVO was observed for all *n*-alkanes except for *n*-C_38_, *n*-C_39_ and *n*-C_40_. On the other hand, the only *n*-alkane that was not statistically different between EVOO and SFO was *n*-C_47_ (Figure 6).

As can be observed in Figure 6A the higher statistical significance in the level of alkanes between AVO and EVOO samples was detected for *n*-C_25_ and *n*-C_29_ with *p*-values of 4.7996e-11 and 7.0023e-11, respectively. *n*-C_26_ was also significantly lower in AVO compared to EVOO. However, it was not taken into account because of the very low average level in AVO (<0.6%) and the *n*-C_29_/*n*-C_25_ ratio was further assessed for revealing AVO in EVOO samples. On the other hand, when considering SFO, the higher statistical significance compared to EVOO was observed for *n*-C_25_ (*p*-value= 2.1716 × 10^−17^), *n*-C_26_ (*p*-value = 9.3622 × 10^−16^)_,_ *n*-C_29_ (*p*-value = 1.4317 × 10^−24^) and *n*-C_31_ (*p*-value = 1.5258 × 10^−13^) (Figure 6B), while long-chain *n*-alkanes (*n*-C_36,_
*n*-C_37,_
*n*-C_38,_
*n*-C_40_) were not considered because of their absence in EVOO accompanied to low levels (<1%) in SFO. Accordingly, two indices based on the ratio (*n*-C_29_ + *n*-C_31_)/(*n*-C_25_ + *n*-C_26_) for SFO and *n*-C_29_/*n*-C_25_ for AVO, were proposed. Finally, for checking the feasibility of the proposed indices, their theoretical changes in different mixtures of EVOO with SFO and AVO at percentages of 2, 5, 10, 30 and 50%, were calculated considering *n*-alkanes compositions of pure oils. Ratios are expressed as minimum and maximum values considering the data set of oil samples analyzed through the study within each oil category. As can be observed in Table 3, the proposed indices allowed to reveal the admixture in EVOO of 2% of SFO and 5% of AVO.

Figure 7 reports from top to bottom the *n*-alkane profile (HPLC-GC-FID traces) of a pure SFO and EVOO and of the same EVOO admixed with 2, 10 and 50% of SFO. As a confirmation of the potential of the index selected for revealing admixtures of a little percentage of SFO in EVOO, the value calculated for the real mixtures were in the predicted ranges.

## 3. Materials and Methods

### 3.1. Reagents and Standards

All solvents were purchased from Sigma–Aldrich (Milan, Italy). Ultra-pure water was obtained with a Milli-Q purification system (Millipore, Bedford, MA, USA). All glassware was carefully washed and rinsed with clean solvents (acetone and hexane) before use. The stock standard mixture *n*-C_10_-C_40_ (50 mg/L each in cyclohexane), used to check GC performance, single *n*-alkane standards (*n*-C_23_, *n*-C_25_, *n*-C_27_, *n*-C_29_, *n*-C_31_, *n*-C_33_, *n*-C_35_) used for calibration and recovery tests, as well as internal standards *n*-C_20_ and *n*-C_40,_ were from Sigma–Aldrich (Milan, Italy). Single *n*-alkane solutions were prepared in toluene (5 mg/mL). A stock standard solution in hexane, containing all the seven *n*-alkanes (only even carbon number) from *n*-C_23_ to *n*-C_35_, was prepared by mixing different amounts of each *n*-alkane in toluene and contained 14.8 mg/mL of *n*-C_23_, 23.5 mg/mL of *n*-C_25_, 71.5 mg/mL of *n*-C_27_, 147.5 mg/kg of *n*-C_29_, 56.3 mg/kg of *n*-C_31_, 14.3 mg/mL of *n*-C_33_ and 6.1 mg/mL of *n*-C_35_. The six standard solutions used for calibration were obtained by diluting the stock standard solution with *n*-hexane as follows: 1:2.5, 1:5, 1:12.5, 1:25, 1:50 and 1:250 (*v*/*v*).

The working internal standard solution consisted of *n*-C_20_ and *n*-C_40_ at 0.30 mg/mL in toluene.

### 3.2. Samples

Six vegetable oils of known origin (directly furnished by the producers) were analyzed with the validated offline SPE-GC-FID method to compare their *n*-alkanes profiles. They included: 1 EVOO, 1 refined OO, 1 refined SFO oil, 1 virgin AVO, 1 refined HZO, and 1 refined PLO. Later 16 EVOO samples (declared 100% from Italian olives), 13 refined SFO, and 8 virgin AVO, of different brands, were randomly purchased from the Italian market and analyzed using online HPLC-GC-FID.

### 3.3. Offline SPE-GC-FID

The starting point for method development was the method proposed for mineral oil determination in vegetable oils by Moret et al. (2011) [16].

#### 3.3.1. SPE

Silver silica (10%) was prepared in the laboratory following the procedure described by Moret et al. (2012) [19]. In brief, a solution, obtained by dissolving 3 g of silver nitrate in 4 mL of Milli-Q water (previously washed with hexane), was added drop by drop (while mixing) to 30 g of silica gel, previously heated overnight at 400 °C. Afterward, the silver silica phase was blended for 30 min in a rotary evaporator without applying the vacuum, protecting the sorbent from the light with an aluminum foil. After a 12-h rest, the silver silica was dried overnight at 75 °C and stored (wrapped with aluminum foil) in a desiccator until use. A 6-mL glass cartridge with a bottom frit of glass fiber (Macherey-Nagel, Chromabond, Düren, Germany) was filled with 1 g of the dried silver silica prepared in the laboratory and positioned in the vacuum manifold (VacMaster-10, Biotage, IST, Stepbio, Bologna, Italy). After adding 2.5 mL of distilled *n*-hexane (maintaining the valve closed), the slurry was mixed with the tip of a Pasteur pipette to release bubbles and left to settle by gentle vibration. Finally, the solvent was drained off and the cartridge was conditioned with 5 mL of *n*-hexane.

Following the procedure described by Moret et al. (2011) [16], 1 g of oil sample, added with 10 µL of the internal standard solution, was diluted to volume with *n*-hexane into a 2 mL volumetric flask and 250 µL of the solution were loaded onto the packed silver silica cartridge. The first 1 mL was discarded, while the following 2 mL fraction containing the *n*-alkanes was collected in a cone-shaped vial, concentrated to about 700 µL (under a nitrogen stream), transferred to an autosampler vial and injected in large volume (50 µL) into the GC-FID system.

#### 3.3.2. GC-FID Analysis

The gas chromatograph was a GC 7890A system (Agilent Technologies, Santa Clara, CA, USA) equipped with an autosampler 7693 (Agilent Technologies) and an on-column inlet. Large volume injection was performed through the on-column injector using the retention gap technique: 50 µL of the sample were injected into a 5 m × 0.53 mm i.d. retention gap (uncoated pre-column), at a controlled injection rate (300 µL/min). The retention gap was connected by a press-fit connector to a 10 m × 0.25 mm i.d. separation column of 0.15 µm of film thickness, coated with cross-linked PS-255 (1% vinyl, 99% methyl polysiloxane) from Mega (Milan, Italy). The oven temperature was programmed from 65 °C (isotherm for 4 min) to 350 °C at 20 °C/min and maintained at 350 °C for 4 min. The carrier gas (helium) flow rate was set at 8 mL/min (constant flow). The FID detector was heated at 350 °C. H2, air and make-up (He) flows were set at 35, 400 and 25 mL/min, respectively. Data were acquired and processed by the ChemStation software (version B.02.01). The *n*-alkanes areas were determined by manual integration of every single peak. Quantification was performed by internal standard, *n*-C_20_.

#### 3.3.3. Linearity

The linearity of the method was tested in a solvent (*n*-hexane). A six-point calibration curve was prepared by properly diluting the stock standard solution containing seven even-numbered *n*-alkanes from *n*-C_23_ to *n*-C_35_ (Table 1). The linearity range tested for every single *n*-alkane was different to better represent the different concentrations at which they are generally present in edible oils and is reported in Table 1. Referring to *n*-C_35_ as the least abundant one and to *n*-C_29_ as the most abundant one, concentrations tested ranged from 0.02 to 2.4 mg/L (corresponding to 0.05–4.9 mg/kg of oil) for *n*-C_35_ and from 0.6 to 59.0 mg/L (corresponding to 1.2–118.0 mg/kg of oil) for *n*-C_29_. Each calibration point was prepared in triplicate and analyzed twice.

#### 3.3.4. Recovery and Repeatability

For recovery tests, an EVOO sample was deprived of endogenous *n*-alkanes and spiked (in triplicate) with different amounts of the stock *n*-alkane solution.

To obtain such oil, 1 g of a selected EVOO (diluted to 2 mL with *n*-hexane) was loaded on a glass column packed with 10 g of activated silica gel (previously washed with dichloromethane and conditioned with *n*-hexane). The *n*-alkanes fraction was eluted with 25 mL of *n*-hexane and discarded. The following 30 mL fraction containing the oil without *n*-alkanes was eluted with dichloromethane and placed in a round-bottomed flask and taken to dryness with a rotatory evaporator. To collect a sufficient amount of clean oil the entire procedure was repeated five times.

Finally, three different 0.5 g aliquots of the cleaned oil were spiked (in triplicate) with three different amounts (7, 30, and 140 µL) of the even *n*-alkane stock solution from *n*-C_23_ to *n*-C_33_ (the same used for calibration), diluted to 1 mL with *n*-hexane and subjected to the SPE-GC-FID procedure. Recoveries were calculated as the percentage ratio of the area obtained from the spiked sample (detracted from the residual *n*-alkanes in the non-spiked sample) and the expected area corresponding to the added amount extrapolated from the calibration line.

Repeatability tests were performed on a native EVOO which underwent the entire procedure six times.

### 3.4. Online HPLC-GC-FID Method

The online HPLC-GC-FID method was the same used for mineral oil determination [16] with some little modifications.

#### 3.4.1. Sample Preparation

One hundred milligrams of the oil sample was weighed into a 1 mL volumetric flask, added with 10 µL of the internal standard (*n*-C_20_) at 0.3 mg/mL, and taken to volume with *n*-hexane. The sample was transferred to an autosampler vial and 50 µL (corresponding to 5 mg of oil) was injected into the online HPLC-GC-FID system. For the AVO, having higher endogenous *n*-alkane content, the amount injected was reduced to 20 µL.

#### 3.4.2. Analytical Determination

HPLC-GC-FID analysis was performed with an online coupled LC-GC 9000 system (Brechbühler, Zurich, Switzerland), consisting of a PAL LHS2-xt Combi PAL autosampler (Zwingen, Switzerland), a Phoenix 40 three syringe LC pump equipped with four switching valves (injection, backflush, transfer, and additional valve) and UV/VIS, UV-2070 Plus detector (Jasco, Japan). Chromatographic conditions were the same as used for mineral oil determination in vegetable oils [17], with the difference that only the channel used for the analysis of MOSH was in use, *n*-C_20_ was used as internal standard and a lower sample amount (corresponding to 5 or 2 mg of oil sample) was injected into the HPLC-GC-FID apparatus.

The HPLC column was a 25 cm × 2.1 mm i.d. packed with LiChrospher Si 60, 5 µm (DGB, Schlossboeckelheim, Germany). The GC was a Trace 1310 series from Thermo Scientific (Milan, Italy). The transfer of the LC fraction into the GC was performed through the Y-interface using the retention gap technique and exploiting partially concurrent eluent evaporation [20]. A 10 m × 0.53 mm i.d. uncoated and deactivated retention gap was followed by a steel T-piece union connected to the solvent vapor exit (SVE) and a 10 m × 0.25 mm i.d. separation column coated with 0.15 µm film of PS-255 (1% vinyl, 99% methyl polysiloxane). Both the retention gap and the column were from Mega (Milan, Italy). Hydrogen was used as carrier gas at a constant pressure of 60 kPa. During the transfer, the pressure increased to 90 kPa. The oven temperature was programmed at 20 °C/min from 50 to 350 °C. FID and SVE were heated at 360 and 140 °C, respectively. Data were acquired and processed by the Chromeleon 7 software (version 7.2.7).

### 3.5. Statistical Analysis

The filtered and normalized data sets were imported into the Metaboanalyst software 5.0 for statistical analysis and PCA. All data were expressed by means ± SD. The statistical significance was tested using Student’s *t*-test. Differences were considered to be significant when *p*-values were <0.05. The threshold for log^2^ fold-change was considered at a min 2-fold difference according to the formula M1/M2 > 2 or M1/M2 < 1/2.

## 4. Conclusions

The rapid offline SPE-GC-FID method proposed for endogenous *n*-alkanes determination in vegetable oils demonstrated good performance characteristics and gave results equivalent to those obtained with online HPLC-GC-FID, without requiring expensive or complex instrumentation. Preliminary results proved that the *n*-alkane profile could be a powerful tool for EVOO purity assessment, able to reveal the addition of small percentages of SFO or AVO in EVOO. Specifically, based on the available data, two powerful indices able to differentiate 2% of SFO in EVOO and 5% of AVO in EVOO, have been proposed for the first time. Further studies analyzing a larger set of pure EVOO and possible adulterants are needed to identify new indices as well as to assess more reliable intervals for the indices here proposed for EVOO authenticity control.

## Figures and Tables

**Figure 1 molecules-28-04393-f001:**
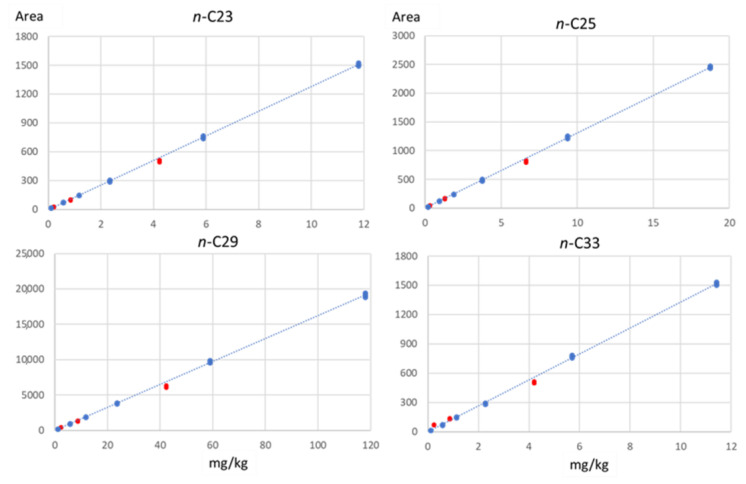
Calibration lines of some selected *n*-alkanes and recovery data at three different fortification levels (red points).

**Figure 2 molecules-28-04393-f002:**
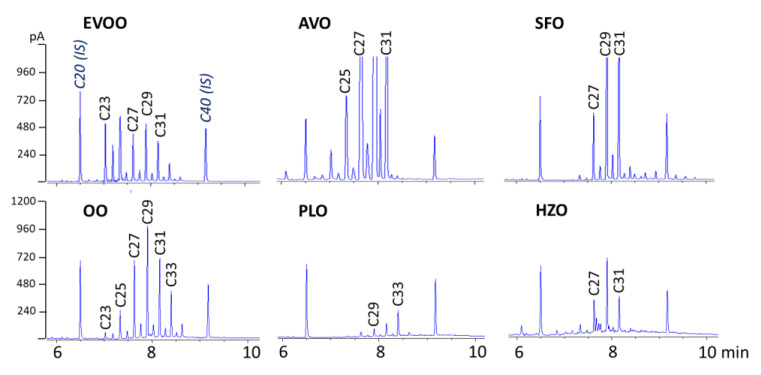
SPE-GC-FID chromatograms of different vegetable oils of certain origin.

**Figure 3 molecules-28-04393-f003:**
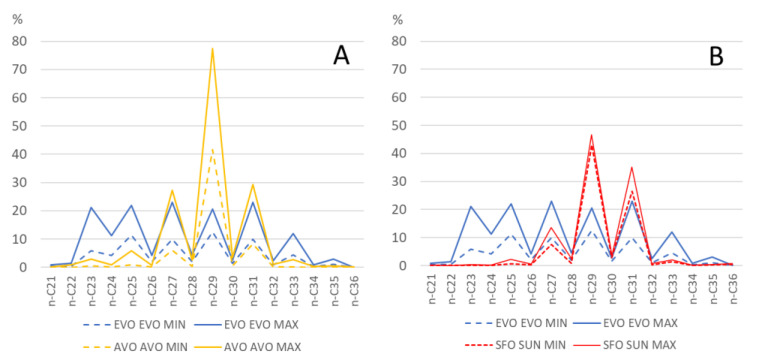
*n*-Alkane percentages (dashed and solid lines indicate minimum and maximum values, respectively). (**A**) EVOO compared to AVO; (**B**) EVOO compared to SFO.

**Figure 4 molecules-28-04393-f004:**
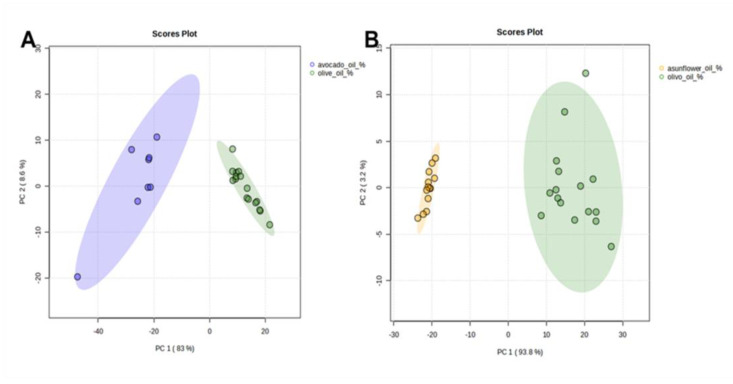
Principal component analysis (PCA) score plot of alkane profile variation for (**A**) avocado (blue dots) and olive oil (green dots) samples; (**B**) for sunflower oil (yellow dots) and olive oil (green dots) samples.

**Figure 5 molecules-28-04393-f005:**
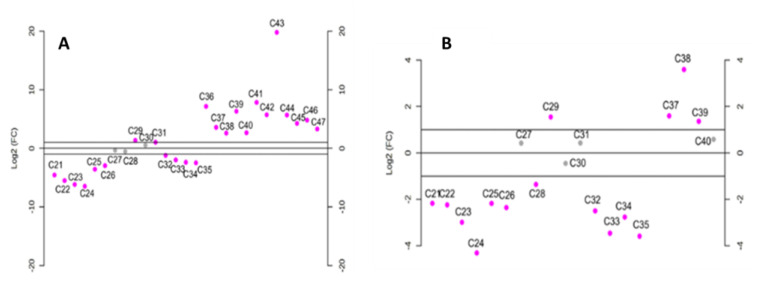
Average log^2^-fold-change (Log^2^FC) for each *n*-alkane in EVOO compared to SFO (**A**) and AVO (**B**).

**Figure 6 molecules-28-04393-f006:**
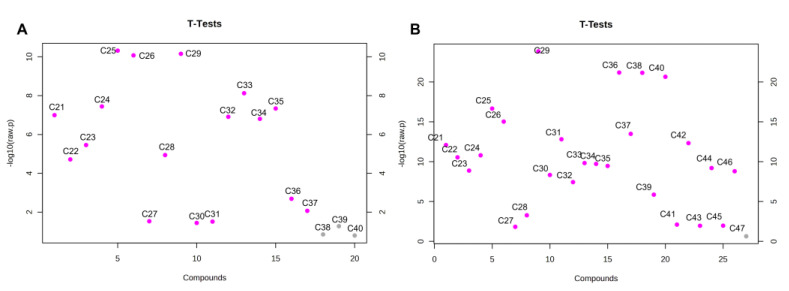
Statistical significance in the mean level of *n*-alkanes between EVOO and AVO (**A**) or SFO (**B**) samples. Significant differences are marked with pink dot (*p* < 0.05).

**Figure 7 molecules-28-04393-f007:**
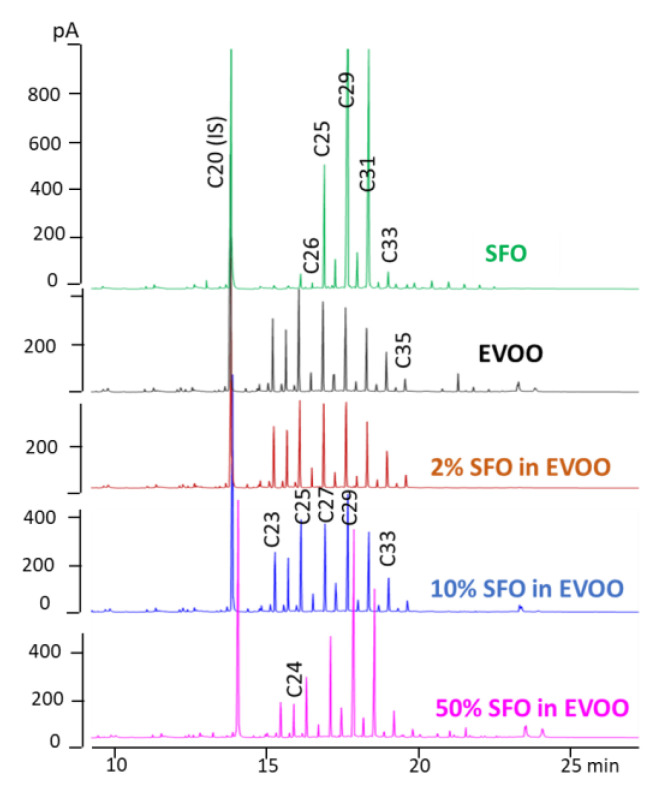
Online HPLC-GC-FID chromatograms of pure SFO and EVOO and of their admixtures (2, 10 and 50% of SFO in EVOO) prepared in the laboratory.

**Table 1 molecules-28-04393-t001:** Linearity range, equation of the regression line, and coefficient of determination (R^2^).

	Linearity Range Tested (mg/kg)	Equation of Regression Line	R^2^
*n*-C_23_	0.12–11.8	y = 128.24x − 8.2644	0.999
*n*-C_25_	0.19–18.8	y = 131.26x − 6.1938	0.999
*n*-C_27_	0.57–57.2	y = 135.52x − 11.739	0.999
*n*-C_29_	1.18–118.0	y = 162.32x − 16.838	0.999
*n*-C_31_	0.45–45.04	y = 135.6x − 16.571	0.999
*n*-C_33_	0.11–11.42	y = 133.63x − 7.1812	0.999
*n*-C_35_	0.05–4.89	y = 132.39x − 3.3442	0.999

**Table 2 molecules-28-04393-t002:** Selected *n*-alkane composition (mg/kg) and total *n*-alkanes in the range *n*-C_21_–C_47_.

	*n*-C_23_	*n*-C_25_	*n*-C_27_	*n*-C_29_	*n*-C_31_	*n*-C_33_	TOT
EVOO1	18.38	16.41	10.05	10.92	8.70	3.88	87
EVOO2	9.20	10.86	7.60	12.07	9.97	4.28	67
EVOO3	4.34	16.42	17.14	12.14	8.33	4.15	75
EVOO4	4.69	8.53	7.49	14.47	17.36	9.07	75
EVOO5	6.52	9.92	9.76	13.97	13.20	5.96	73
EVOO6	10.79	15.75	8.76	11.39	9.26	4.34	77
EVOO7	5.91	10.40	10.28	13.75	11.54	5.06	69
EVOO8	5.55	9.57	8.64	14.40	13.23	5.94	70
EVOO9	6.72	10.20	9.31	13.81	13.83	5.96	76
EVOO10	8.46	12.24	10.43	9.94	7.80	4.81	69
EVOO11	8.84	11.86	10.10	11.73	9.10	3.46	68
EVOO12	15.75	15.26	11.71	12.81	10.53	4.67	89
EVOO13	4.94	8.69	9.28	12.01	9.47	3.40	58
EVOO14	3.04	7.26	9.37	8.57	6.55	3.00	46
EVOO15	8.09	9.90	7.01	8.57	7.36	3.32	56
EVOO16	4.11	7.20	5.18	8.32	8.37	3.04	45
Mean	7.83	11.28	9.51	11.80	10.29	4.65	60
AVO1	0.43	1.46	10.59	33.35	23.43	2.14	80
AVO2	8.20	21.99	113.01	232.86	121.12	0.74	511
AVO3	1.06	4.10	10.93	138.27	14.86	1.49	179
AVO4	2.54	6.20	27.87	55.06	19.60	0.44	117
AVO6	2.06	13.51	91.04	169.34	41.94	0.54	334
AVO7	3.16	3.73	73.97	227.67	129.29	0.26	447
AVO8	11.71	23.04	81.06	183.11	71.11	1.88	391
AVO9	8.81	20.34	91.49	208.64	114.50	0.89	458
Mean	4.75	11.80	62.50	156.04	66.98	1.05	304
SFO 1	0.32	2.47	16.46	59.99	37.88	1.70	130
SFO 2	0.41	2.99	18.46	62.04	38.35	1.84	135
SFO 3	0.12	1.50	13.51	53.99	35.00	1.96	118
SFO 4	0.10	2.17	24.83	86.15	59.86	3.87	200
SFO 5	0.15	1.56	14.15	55.19	38.98	2.27	126
SFO 6	0.27	1.88	12.86	43.04	24.93	1.60	94
SFO 7	0.09	0.88	8.52	40.07	31.54	1.82	93
SFO 8	0.40	2.61	15.89	65.27	44.14	2.69	145
SFO 9	0.17	1.57	11.92	46.25	32.51	1.73	103
SFO 10	0.18	1.30	10.73	41.70	33.02	1.56	97
SFO 11	0.11	1.53	16.81	66.40	47.31	2.79	149
SFO 12	0.05	0.64	7.05	33.55	26.99	1.37	77
SFO 13	0.04	0.67	7.69	47.31	33.63	1.78	102
Mean	0.18	1.68	13.76	53.92	37.24	2.07	109

**Table 3 molecules-28-04393-t003:** Theoretical indices in pure EVOO and mixture containing 2, 5, 10, 30 and 50% of SFO or AVO.

		% Adulterant in EVOO					Pure EVOO
Oil	Index	2	5	10	30	50	
		Min/Max	Min/Max	Min/Max	Min/Max	Min/Max	Min/Max
SFO	*(n-C* _29_ *+ n-C* _31_ *)/* *(n-C* _25_ *+ n-C* _26_ *)*	1.73/1.78	1.83/1.95	2.00/2.24	2.88/3.80	4.35/6.45	1.67/1.68
AVO	*n-C* _29_ */n-C* _25_	1.00/1.18	1.11/1.30	1.29/1.50	2.20/2.61	3.52/4.47	0.94/1.11

Values are expressed as minimum (min) and maximum (max) levels in EVOO (*n* = 16), AVO (*n* = 9), and SFO (*n* = 13) samples. Bold figures indicate the percentage of adulterant that can be discriminated from pure EVOO.

## Data Availability

Data are contained within the article.

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
