# Peer review of "Endogenous n-Alkanes in Vegetable Oils: Validation of a Rapid Offline SPE-GC-FID Method, Comparison with Online LC-GC-FID and Potential for Olive Oil Quality Control"

_molecules, 2023, doi:10.3390/molecules28114393_

Round 1
Reviewer 1 Report
Dear Authors,
I read your paper carefully but it is not clearly presented. Please indicate what should be a general n-alkane profile for EVOO. If the total n-alkanes of other oils are sometimes same as of EVOO, then what should be the main criteria for differentiation.
The caption of Table 1 is linearity. It is not suitable.
How the calibration standards were prepared?
When all methods employed for sample preparation are previously published, what is the novelty of this work?
Explain off line SPE with more clarity. How interferences from other hydrocarbons were avoided?
What was the criteria for selection of spiked concentrations in recovery experiments?
Show Y-axis of the chromatograms.
Please add chromatograms of EVOO without and with adulteration.
Overall, this paper needs better organization and presentation. Additional sub sections can be added for better flow and understanding of the text.
Author Response
Dear Authors,
I read your paper carefully but it is not clearly presented.
We thank the reviewer for taking the time to review our paper, which gave us the opportunity to improve it. The text has been revised based on the reviewer's suggestions/comments, creating additional subsections, and trying to make the text clearer. We have also slightly changed the title to make the content of the article clearer.
Please indicate what should be a general n-alkane profile for EVOO.
The general n-alkane profile of EVOO compared to SFO and AVO, can be appreciated in fig. 3. A paragraph elucidating the EVOO profile compared to that of SFO and AVO has been added soon after fig. 3. Furthermore, an additional figure (fig. 6) showing the different profile of EVOO, SFO and their mixtures has been added in section 2.
If the total n-alkanes of other oils are sometimes same as of EVOO, then what should be the main criteria for differentiation.
Further comments have been added in the text to better explain what is the potential of quali-quantitative composition of n-alkanes and what are the main criteria for olive oil differentiation (see section 2.6). About this, the n-C29/n-C24 ratio was proposed in the original manuscript. Nevertheless, during the revision we realized that there was a formula error in the Excel file (we are mortified by this mistake) and that, different from what stated, this ratio allowed for detecting only 30% of SFO and AVO in EVOO. Therefore, based on the statistical evaluation, two more powerful indices able to reveal the addition of 2% of SFO and 5% of AVO in EVOO have been investigated and proposed, as added in the text (see paragraph 2.6.2). We apologize for the error and thank the reviewer for giving us the opportunity to discover and correct it.
The caption of Table 1 is linearity. It is not suitable.
The caption of figure 1 has been modified as requested.
How the calibration standards were prepared?
A sentence explaining how the calibration standards were prepared has been added in section 3.1.
When all methods employed for sample preparation are previously published, what is the novelty of this work?
As reported in the text, the method proposed for the off-line determination of n-alkanes in vegetable oils is derived from a method previously proposed by some of the same authors for MOSH determination in the same matrix. Nevertheless, in the present case, the method, which required some modifications reported in the text, was used for n-alkanes determination and was validated for the first time for these analytes. In addition, the SPE-GC-FID method was compared with the on-line LC-GC-FID technique, obtaining comparable results, and potentiality of n-alkane profile for revealing admixture of cheaper oil (avocado oil and sunflower oil) with extra virgin olive oil have been investigated for the first time. All this represents a novelty and is now better underlined in the introduction.
Explain off line SPE with more clarity. How interferences from other hydrocarbons were avoided?
Section 2.1 on the optimization of the off-line SPE-GC-FID procedure has been modified to clarify its content. In particular, the explanation relating to the possible interference due to mineral oil hydrocarbons has been implemented.
What was the criteria for selection of spiked concentrations in recovery experiments?
Since different n-alkanes with different concentration ranges are present in the oils considered, we tried to reproduce these differences and the amount added approximately covered the range usually found in EVOO. A comment on this has been added in the text (section 2.3).
Show Y-axis of the chromatograms.
Y-axis have been added in figure 1.
Please add chromatograms of EVOO without and with adulteration.
A figure of an EVOO adulterated with 2%, 10% and 50% of SFO, together with the corresponding pure EVOO and SFO has been added in the text.
Overall, this paper needs better organization and presentation. Additional sub sections can be added for better flow and understanding of the text.
The text has been revised, creating additional subsections to make it clearer.
Reviewer 2 Report
The research's main question addressed quantitative determination of aliphatic hydrocarbons in vegetable oils by SPE-GC-FID and HPLC-GC-FID. The topic is relevant in the field. Compared with other published material, it add to method validation and comparison with on-line LC-GC-FID and potential for olive oil quality control. The conclusions are consistent with the evidence and arguments presented and they address the main question posed. The references are appropriate.
Several specific comments,
-the quality is good but in Figs. 1-2, the labeling of the axes is missing.
- On the page 11, in line 383 should be: LiChrospher Si 60, 5 µm
Author Response
The research's main question addressed quantitative determination of aliphatic hydrocarbons in vegetable oils by SPE-GC-FID and HPLC-GC-FID. The topic is relevant in the field. Compared with other published material, it add to method validation and comparison with on-line LC-GC-FID and potential for olive oil quality control. The conclusions are consistent with the evidence and arguments presented and they address the main question posed. The references are appropriate.
The Authors thank the reviewer for the time spent and positive comments. The manuscript has been revised according to the suggestions.
Several specific comments,
-the quality is good but in Figs. 1-2, the labeling of the axes is missing.
Figures 1-2 have been modified accordingly.
- On the page 11, in line 383 should be: LiChrospher Si 60, 5 µm
The text has been modified accordingly.
Round 2
Reviewer 1 Report
Authors have tried to address my comments.